# Sea Anemone-Derived Toxin Avd3i Inhibited Sodium Channel Nav1.4

**DOI:** 10.3390/toxins17090461

**Published:** 2025-09-13

**Authors:** Jiaxin Gao, Guohao Liu, Yan Liu, Dezhao Zhang, Qinyi He, Qiong Liao, Canwei Du

**Affiliations:** 1School of Life and Health Sciences, Hunan University of Science and Technology, Xiangtan 411201, China; 2The National & Local Joint Engineering Laboratory of Animal Peptide Drug Development, College of Life Sciences, Hunan Normal University, Changsha 410006, China; 3Hunan Provincial Key Laboratory of Animal Intestinal Function and Regulation, College of Life Sciences, Hunan Normal University, Changsha 410006, China

**Keywords:** Nav1.4, Avd3i, inhibition, purification

## Abstract

Ion channels regulate ion transport across cell or organelle membranes, playing an important role in various biological processes. Sodium channel Nav1.4 is critical to initiating and propagating action potentials in skeletal muscles, and its dysfunction is associated with a variety of diseases, such as non-dystrophic myotonias. In this study, U-actitoxin-Avd3i (Avd3i), a Kunitz-type toxin derived from *Anemonia viridis*, was expressed in prokaryotic systems and was subsequently purified via high-pressure liquid chromatography. Patch clamp recording showed that Avd3i inhibited Nav1.4 in a concentration-dependent manner, with an IC_50_ of 25.43 μM. However, the toxin exerted no inhibitory activity in Nav1.5/Nav1.7 channels or Kv1.1/Kv1.3/Kv1.4/Kv4.2 potassium channels. Our study found that the sea anemone-derived toxin Avd3i inhibited sodium channel Nav1.4, providing a novel molecule that can act on the channel.

## 1. Introduction

Ion channels are a class of highly specific protein complexes on the cell membrane; they are distributed throughout the cells of an organism, such as immune cells and nerve cells [1,2]. Ion channels can regulate ion flow across cell membranes, and they play a function in signal transduction, which is the key to maintaining a normal state and progressing through the different stages of life [1,2]. Ion channels are divided into three categories according to their gating mechanisms: (1) mechanosensitive channels, which are sensitive to the mechanical stress of the cell membrane, open or close by sensing physical deformation of the membrane [3]; (2) ligand-gated channels, the activity of which is regulated by specific chemical signals; and (3) voltage-gated channels, such as Nav, Cav, and Kv channels, in which the S4 transmembrane segment carries periodic positively charged residues and senses changes in membrane potential through spiral rotation [4]. Ion channels play a crucial role in human physiological functions. For instance, voltage-gated sodium channels drive a rise in the action potential in excitable cells [5]. Voltage-gated sodium channels are transmembrane proteins composed of α-subunit and auxiliary β-subunits. Its α-subunits form the core structure of the ion channel, including four homologous domains (DI~DIV). Each domain contains six transmembrane helices (S1–S6), of which S1–S4 constitute a voltage sensor responding to the change in the membrane potential and S5–S6 participate in the formation of ion conduction pores, offering ions a pathway across cell membranes [2]. β-subunits play an auxiliary role by helping to locate the channel on the cell membrane or modulating its functional properties [6]. There are nine different α-subunits (NaV1.1–NaV1.9) that are known to differentiate in tissue distribution and function, and their mutations can cause nervous system, cardiovascular system, and musculoskeletal system dysfunctions and diseases [7]. Nav1.4 is mainly expressed in skeletal muscles; it functions by evoking the action potentials, thereby triggering and regulating muscle contractions [2]. Normally, Nav1.4 remains closed in a resting state. However, when the cell membrane is stimulated, the channel opens up rapidly, allowing for sodium ions to enter the cell, resulting in the depolarization of the cell membrane and thus initiating an action potential [1]. The generation and propagation of action potentials are essential for the normal contraction and relaxation of muscles. However, Nav1.4 dysfunction can lead to changes in muscle excitability, resulting in a rise in various muscle diseases, such as hypokalemic periodic paralysis, hypercalcemic periodic paralysis, and congenital myotonic paralysis [2,7,8].

Sea anemones are typical representatives of the phylum Cnidaria, whose venom, secreted by their nematocysts, is a mixture of peptides, proteins, and non-protein compounds [9,10]. A recent review summarized that 236 toxic peptides have been isolated from more than 45 species of anemone, and over 60 protein toxins were obtained in their natural conformation [5]. These toxins could be classified into voltage-gated sodium channel toxins (NaTxs), voltage-gated potassium channel toxins (KTxs), acid-sensitive channel toxins (APETx-like peptides), cytolysins, Kunitz-type protease inhibitors, and toxins with phospholipase A2 activity [10]. Among them, NaTxs can be divided into one to four types according to their structures, and they interfere with the channel inactivation process by differentially binding to the voltage sensor (VSD-IV) of Nav channel domain IV, exhibiting different selectivity for insect and mammalian Nav subtypes. For example, ATX-II, a neurotoxin (NaTxs) obtained from *Anemonia sulcate*, displays an EC_50_ of 49.05 nM for NaV1.5 but an EC_50_ of 109.49 nM for NaV1.4 [5].

Kunitz-type peptides are a class of peptides with specific structures and functions that are formed in the long-term evolutionary process; they are equipped with the functions of inhibiting protease activity and regulating ion channels. At present, Kunitz-type peptides have been discovered and screened in many animals, plants, and microorganisms [11]. Anemone venom is rich in a variety of peptides, including toxins that act on sodium ion channels, such as ATX-II [5] and Cgtx-II [10]. In this study, we explored Avd3i from *A. viridis*, which has a high degree of sequence conservation compared to Kunitz-type peptides, and we obtained its spatial structure via molecular modeling. To further analyze the function of Avd3i, we obtained the peptide by performing prokaryotic expression and purification, and patch clamp recording was carried out to assess its activities on some ion channels, specifically exploring its interaction with Nav1.4 channels. The results showed that it exerted a significant inhibitory effect on Nav1.4, and the inhibitory activity was in a peptide concentration-dependent manner. The possible interaction mode between Avd3i and Nav1.4 was analyzed via random molecular docking.

## 2. Results

### 2.1. Structural Analysis of Avd3i

The peptide U-actitoxin-Avd3i (Avd3i) contains 59 residues with a molecular weight of 6683.57 Da, and it typically forms three pairs of highly conserved intramolecular disulfide bonds. As shown in Figure 1A, Avd3i shared high homology with Kunitz-type peptides derived from sea anemones; it had 90%, 86%, and 73% similarity with Avd3d, Avd3b [12], and APEKTx1 [13], respectively. Textilinin-1 is a typical Kunitz-type peptide extracted from the Eastern brown snake [14]. Since its spatial structure was elucidated via X-ray diffraction (PDB ID: 3BYB) with a resolution of 1.64 Å [15] and displayed 58% similarity with Avd3i, Textilinin-1 was used as the template to build the predicted structure of Avd3i. The spatial structure of Avd3i was obtained using SwissModel homology modeling (https://swissmodel.expasy.org/), with a Global Model Quality Estimate of 0.72 (Figure 1B). The structure of Avd3i was also predicted using AlphaFold3, showing that Avd3i contained two stable anti-β parallel sheets and one short α-helical structure (Figure 1C).

### 2.2. Prokaryotic Expression and Purification of Avd3i

We used prokaryotic expression in *Escherichia coli* (BL21) to obtain peptide Avd3i. As shown in Figure 2A, the sequence encoding Avd3i was inserted into vector pET-32a (+) with His-tag (6 × His) and followed by nucleic acid encoding TEV (Tobacco Etch Virus) protease recognition sites. To purify the recombinant protein, we performed nickel column affinity chromatography, and peptide Avd3i was released by TEV protease (Figure 2B). Peptide Avd3i was further purified using high-performance reverse liquid chromatography (RP-HPLC) (Figure 2C). To identify peptide Avd3i according to its prokaryotic expression, MALDI-TOF (matrix-assisted laser desorption tandem time of flight) mass spectrometry was carried out. Mass spectrometry showed that the molecular weight of recombinant Avd3i was 7010 Da, which was 6 Da less than its theoretical molecular weight, suggesting that peptide Avd3i may build three pairs of intramolecular disulfide bonds (Figure 2D). The above results showed that peptide Avd3i was obtained through prokaryotic expression and RP-HPLC purification.

### 2.3. Peptide Avd3i Had No Effect on Some Ion Channels

Kunitz-type peptides are a class of multifunctional peptides that not only act as protease inhibitors to regulate multiple protease activities but also inhibit ion channels, such as ShPI-1, which can inhibit various serine proteases and voltage-gated potassium ion channels [16]. Peptides Avd3d, Ael3a, and Avd3b belong to bifunctional Kunitz-type peptides, with high homology to Avd3i, and they exerted a high affinity to potassium channels. Avd3d and Avd3b showed an inhibitory ability on the Kv1.2 channel, with IC_50_ values of 1.3 μM and 2.8 μM, respectively [12,17]. Ael3a is a Kv1.1 channel blocker (IC_50_ of 0.9 nM), and we speculated that it would bind to the channel through a spatial electrostatic interaction, blocking the channel pores in an open state [13]. To assess the effect of peptide Avd3i on voltage-gated potassium channels, patch clamp recording was performed. As shown in Figure 3A–D, 50 μM of Avd3i was added to HEK cells transfected with different channel plasmids; however, Avd3i displayed no effect on potassium channels Kv1.1, Kv1.3, Kv1.4, or Kv4.2. We also assessed the effect of Avd3i on voltage-gated sodium channels Nav1.5 and Nav1.7, but the peptide exerted no function on either channel (Figure 3E,F).

### 2.4. Peptide Avd3i Inhibited Nav1.4 Channel

We continued to assess the effect of peptide Avd3i on the Nav1.4 channel and found that the Nav1.4 channel could be suppressed by the perfusion of 50.0 μM of Avd3i (Figure 4A). The inhibition of Avd3i occurred in a concentration-dependent manner, with an IC_50_ value of 25.43 μM (Figure 4B). Then, random molecular docking was carried out to predict the interaction of Avd3i binding to the Nav1.4 channel. The predicted structure of peptide Avd3i obtained using SWISS model (Figure 1B) was set as the ligand, while the cryo-EM structure of the Nav1.4 channel (PDB ID: 5XSY [6]) was chosen as the receptor. With the performance of random docking, about 2000 poses were obtained. Considering that peptide Avd3i perfused by the extracellular side, four clusters were presumed to be the probable representative poses of Avd3i binding to the Nav1.4 channel (Figure 4C). Specifically, the probable poses showed that α-helix in peptide Avd3i might act on the junction between Nav1.4 DIV-S6 and DIV-S5 (Figure 4D), or the parallel β-sheet in Avd3i may bind to the junction between Nav1.4 DIV-S6 and DIV-S5 (Figure 4E). Other clusters showed that the N-terminal and C-terminal in Avd3i may be inserted into the cavern formed by Nav1.4 DII-S2 and DII-S3 (Figure 4F). Therefore, we found that peptide Avd3i inhibited the Nav1.4 channel.

## 3. Discussion

Ion channels are widely distributed in various cells, regulating cellular electrical activities, and their functions are essential to maintaining normal organism processes. A variety of processes, ranging from daily physical activities such as walking and raising one’s hands to partaking in complex activities such as dancing and athletics, rely on proper functioning of the Nav1.4 channel. Mutations in the Nav1.4 channel cause a range of muscle diseases, such as non-dystrophic myotonia (NDM) [18] and periodic paralysis (PP) [19,20]. Patients with NDM have difficulty relaxing their muscles after contraction, with symptoms such as muscle stiffness, often accompanied by muscle pain, which greatly affects their quality of life [21,22]. Periodic paralyzes are classified into distinct hyperkalemic (hyperPP) and hypokalemic (hypoPP) forms [20]. Patients with PP may experience episodic and permanent muscle weakness, which can lead to paralysis in severe cases [23]. The Nav1.5 channel is related to the regular contraction and relaxation of the heart, and its dysfunction can cause serious, life-threatening heart diseases such as Brugada syndrome and long QT syndrome [24]. TREK-1 is a potassium channel widely expressed across diverse tissues such as sensory neurons in the nervous system and the heart and pulmonary arteries within the cardiovascular system, and it is strongly associated with the development of depression [25,26]. In addition, the calcium channel Cav1.1 could convert the electrical signal of the cell membrane into an intracellular calcium signal, which plays a key role in the excitatory contraction coupling of skeletal muscles [27].

Nature’s almost miraculous evolutionary processes provide precious resources. Many toxic animals, such as spiders, centipedes, scorpions, and snakes, possess various types of venom, including peptide toxins that are usually composed of 20–50 residues, rich in conserved domains, and intramolecular disulfide bonds used to stabilize their unique tertiary structure (e.g., ICK fold or three-finger structure) [28,29]. Some peptide toxins, which can accurately bind to the specific domains of ion channels, interfere with the transmission of nerve signals [30] or the progress of immune responses. These peptides are not only important weapons that animals use to survive and compete in nature but also good tools and potential sources of drugs for ion channel research. For example, ProTx2, a spider toxin isolated from the venom of Peruvian green velvet tarantula, exerted a high affinity to the Nav1.7 channel (IC_50_ = 0.26 nM). Mechanically, ProTx2 can tightly bind to the voltage sensor domain II (VSD2) of the Nav1.7 channel, affecting the conduction of nerve signals and changing physiological processes such as pain perception [31]. LCTx-F2, derived from *Lycosa singoriensis*, can inhibit the ion channels rASIC3 and rASIC1a [32]. IstTx, isolated from *Ixodes scapularis*, selectively inhibits the TREK 1 channel in the K2P channel [33]. The above fruitful and epoch-making examples reveal the great potential of peptide toxins as molecular tools for use in ion channel studies and as candidates for disease-targeted drugs.

Although these peptide toxins in biological venoms may pose to be life-threatening to other organisms, an increasing number of studies have proved that some peptides derived from animal venom are effective candidates for the treatment of diseases [34]. Therefore, the search for natural peptide toxins that can selectively act on ion channels and inhibit their functions has become an important research topic. Up to now, a variety of therapeutic drugs have been developed for Nav1.4 channel diseases, such as mexiletine, tocainide, and flecainide. For instance, mexiletine is an antiarrhythmic drug that binds to DIII S6 and DIV S6 of the sodium channel pore domain to reduce the over-release of the action potentials of muscle fibers by prolonging the refractory period, thereby improving muscle stiffness in patients with NDM [5]. However, it may cause side effects such as indigestion in patients, and the sensitivity of different patients varies greatly [35]. Tocainide is a class IB antiarrhythmic drug that can treat abnormal heart rhythms and improve heart function, but the side effects involve multiple systems and cannot be ignored because of their severity. For example, they can lead to anemia, agranulocytosis, pulmonary fibrosis, interstitial pneumonia, and pulmonary edema, which can be fatal [36]. Although flecainide is a class Ic antiarrhythmic agent and is also effective in some patients with NDM [37], caution should be exercised when using it to avoid adverse effects such as hypotension and heart failure aggravation [38].

A series of Kunitz-type peptides have been found to act on ion channels [39,40]. For example, anemone toxin APEKTx1 specifically inhibited the Kv1.1 channel, with residues Phe13 and Arg15 assumed to play crucial roles in the inhibition of the channel [13]. In this study, we found that peptide Avd3i displayed high homology with the toxin APEKTx1, with a substitution of Pro at residue Phe13 (Figure 1A). Patch clamp recording showed that peptide Avd3i almost lacked inhibitory activity against the Kv1.1 channel (Figure 3A), possibly due to the substitution of residue Phe13. However, this assumption must be verified with a solid exploration, such as the site-directed mutagenesis of peptides. In this study, the peptide Avd3i from *A. viridis* demonstrated no activity against voltage-gated sodium channels (Nav1.5/1.7) and potassium channels (Kv1.1/Kv1.3/Kv1.4/Kv4.2) but displayed a concentration-dependent inhibition of the Nav1.4 channel (IC_50_ = 25.43 μM). Considering that Numbers of Kunitz-type peptides have been reported to inhibit serine proteases, the function of peptide Avd3i acting on serine proteases will be explored in the future. It is well-known that there are various methods for peptide toxins to interact with ion channels. Some peptide toxins can specifically bind to the pore region of the ion channel like a “plug”, blocking the channel and thus preventing the passage of ions. For example, SsTx-4, isolated from the venom of *Scolopendra subspinipes mutilans*, is a potent inhibitor of three inwardly rectifying potassium channels (Kirs) [41,42]. Some peptide toxins alter the gating properties of ion channels so that the opening probability, opening time, and closing time of the channel change. One such example is rpTx1 from *Scolopendra subspinipes mutilans*, which can enter the cell interior and target a structure called the “IFMT” on the sodium channel, leaving the sodium channel in a continuously open state, resulting in continuous cell excitation [43]. Certain peptide toxins do not act directly on ion channels but rather through “indirect routes”. For example, α-Conotoxins (e.g., Vc1.1 and RgIA) do not directly bind to N-type (Cav2.2) calcium channels but indirectly inhibit channel function by binding to G-protein-coupled GABA_B_ receptors, thereby initiating intracellular signaling pathways (which may involve src tyrosine kinase) [44]. Certain peptide toxins, such as ProTx2, may interfere with the interaction between ion channel subunits, affecting their assembly or stability [31]. In this study, we used molecular docking to analyze the probable positions of peptide Avd3i acting on the Nav1.4 channel, providing a reference for studying toxin–channel interactions; these properties differ from those of other Nav1.4 channel inhibitors. We hope that this analysis will shed light on the study of ion channels, especially the Nav1.4 channel.

## 4. Conclusions

A number of peptides, including Kunitz-type peptides, have been found in sea anemones, in which they play a crucial role in physiological processes. In this study, a typical Kunitz-type peptide Avd3i derived from *A. viridis* was explored, with three pairs of intramolecular disulfide bonds. We performed prokaryotic expression and purification to obtain peptide Avd3i. The electrophysiological recording showed that the peptide inhibited the Nav1.4 channel, but it exerted no effect on several other channels. Random molecular docking was carried out to display the probable interaction of Avd3i binding to the Nav1.4 channel. This study found that the peptide Avd3i can suppress the Nav1.4 channel, showcasing a novel molecule that acts on the channel.

## 5. Experimental Procedures and Materials

### 5.1. Sequence Alignment and Homologous Modeling

The amino acid sequence of Avd3i was derived from the cDNA library of *A. viridis* (NCBI Accession: PRJNA260824 ID: 260824) and indexed by the NCBI database [45]. We obtained the peptide sequences of Avd3d, Avd3b, Ael3a, and Textilinin-1 by performing a homologous search of the NR database (https://blast.ncbi.nlm.nih.gov/Blast.cgi, accessed on 20 May 2025) of the NCBI. The Avd3i sequence was used as a query probe, and the E-value threshold was set to 1 × 10^−10^ to screen for sequences with ≥30% homology. MEGA 11 software was used for the multi-sequence alignment, and the ClustalW algorithm was used to generate alignment results and annotate conserved domains. Based on the spatial structure of Textilinin-1 (PDB ID: 23BYB) obtained via X-ray diffraction, we predicted the structure of Avd3i using SwissModel (https://swissmodel.expasy.org/) and AlphaFold3 to obtain a visualized 3D structure.

### 5.2. Recombinant Expression and Purification of Avd3i

To obtain the Avd3i peptide, prokaryotic expression was performed, as outlined previously [33]. Briefly, the sequences encoding the Avd3i amino acid, the TEV protease recognition sequence, and the His tag were inserted into the vector pET-32a (+). The recombinant plasmid was transformed into *E. coli* BL21 (DE3) competent cells (NO. B528419, purchased from Sangon Biotech (Shanghai, China) Co., Ltd.) for isopropyl-beta-D-thiogalactopyranoside (IPTG)-induced expression. The bacterial solution was inoculated into Luria–Bertani liquid medium containing 100 μg/mL of ampicillin and incubated with a shaking speed of 220 rpm at 37 °C for 4 h. When the OD_600_ reached approximately 0.4–0.6, 1 mM of IPTG was added. The solution was then induced with a shaking speed of 100 rpm at 28 °C for 16 h to promote the expression of the fusion protein. The induced bacteria were collected to be crushed for 20 min using a high-pressure homogenizer, and the supernatant was collected via centrifugation at 8000× *g* for 5 min. TEV enzyme (final concentration of 0.1 mg/mL) was added to the eluate product and digested at 16 °C for 12 h. The obtained enzyme digestion solution was purified again using a nickel column to obtain the target protein. The enzyme digest product was verified using 4–12% SDS-PAGE, and the band results were obtained via Coomassie blue staining. The digested samples were purified on a C4 reversed-phase column (4.6 mm × 250 mm, 300A, 3.5 µm) and eluted from 10% to 45% acetonitrile over 40 min using a gradient elution of 0.1% TFA in ddH_2_O (phase A) and acetonitrile (phase B). The elution peaks were then collected and lyophilized using a freeze dryer at −86 °C for 48 h. MALDI-TOF/TOF mass spectrometry was performed to measure the molecular weight, confirming that we had obtained a highly pure peptide Avd3i.

### 5.3. Electrophysiological Recording

Plasmids containing sodium channels or potassium channels were transfected into HEK cells (gifted from Rong’s lab at Hunan Normal University, Changsha, China [32]) using Lipofectamine 2000. After 24 h, whole-cell patch clamp recordings were made at room temperature. Using a drawn borosilicate glass electrode (resistance 3–5 MΩ) filled with intracellular solution, the current of the transfected cells was assessed using the MultiClamp 700B amplifier (Molecular Devices, Sunnyvale, America) and pCLAMP 10 software. We set the clamping voltage to −80 mV and the stimulation voltage to 0 mV. The control (perfusing with extracellular solution) current was recorded first, and the current stimulation was stabilized before adding Avd3i external solution.

The extracellular solution used to record the sodium ion channels included the following components (in mM): 150 NaCl, 2 KCl, 1.5 CaCl_2_, 1 MgCl_2_, and 10 HEPES, and the pH was adjusted to 7.4 with NaOH. To record the potassium ion channel, the extracellular solution contained (in mM) 140 NaCl, 2 KCl, 1.5 CaCl_2_, 1 MgCl_2_, and 10 HEPES, and the pH was adjusted to 7.4 using NaOH. The intracellular solution that recorded the sodium ion channel currents contained 10 mM NaCl, 10 mM EGTA, 135 mM CsCl, and 10 mM HEPES, and the pH was adjusted to 7.3 with CsOH. To record the potassium ion channel currents, the intracellular solution contained 10 mM EGTA, 140 mM KCl, 2.5 mM MgCl_2_, and 10 mM HEPES, and the pH was adjusted to 7.4 using NaOH.

### 5.4. Molecular Docking

The predicted Avd3i structure and the Nav1.4 channel structure model (PDB ID: 5XSY) were used to perform ZDOCK docking in Discovery Studio 2021 software to predict the possible binding conformations and screen for high-scoring models, as outlined previously [46]. The angular step size was set to 6°, 6.0 Å for the RMSD cutoff, and 9.0 Å for the interface cutoff in all ZDOCK dockings. No key junction points were set.

## Figures and Tables

**Figure 1 toxins-17-00461-f001:**
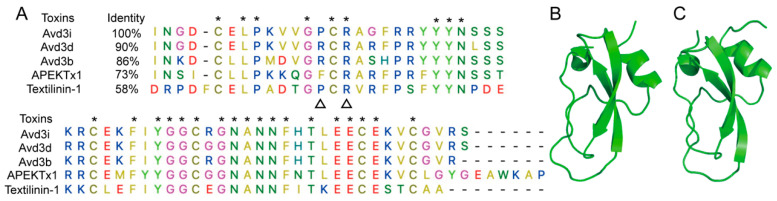
Structure of peptide Avd3i. (**A**) Residue sequences of Avd3i and its alignments with several Kunitz-type peptides. Cysteines and conserved residues are labeled with *, and residues predicted to be important for Kv channel inhibition are labeled with a triangle. (**B**,**C**) Predicted spatial structure of Avd3i obtained using SwissModel (**B**) and AlphaFold3 (**C**).

**Figure 2 toxins-17-00461-f002:**
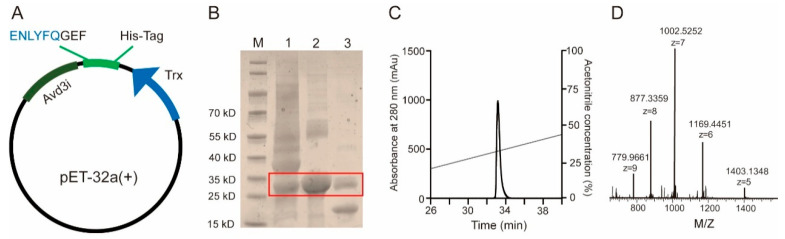
Recombinant expression and purification of peptide Avd3i. (**A**) The sequence encoding peptide Avd3i was inserted into vector pET-32a (+) plasmid. TEV protease recognition site (ENLYFQGEF) and 6 × His tag were labeled. (**B**) The recombinant protein was purified using Ni-NTA column. M, marker; line 1: *E. coli* was induced using IPTG; line 2: recombinant protein after purification using Ni-NTA column; line 3: recombinant protein was cleaved using TEV protease. (**C**) Peptide Avd3i was purified using RP-HPLC. (**D**) Peptide Avd3i was identified using MALDI-TOF, M/Z, mass-to-charge ratio.

**Figure 3 toxins-17-00461-f003:**
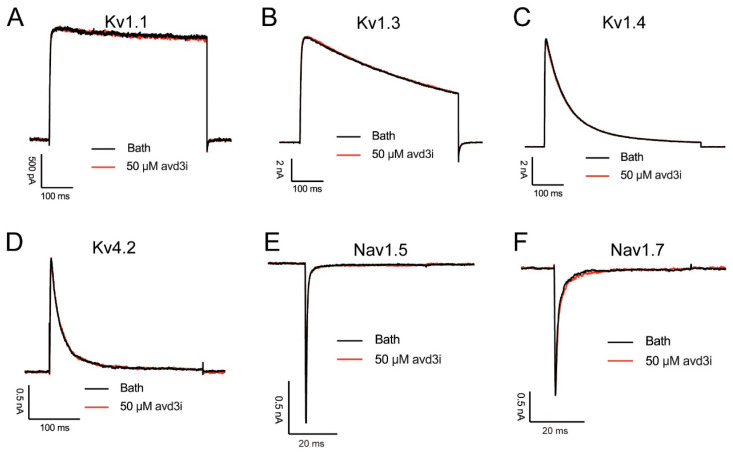
Effect of peptide Avd3i on some channels. (**A**–**D**) Peptide Avd3i did not inhibit the voltage-gated potassium channels Kv1.1 (**A**), Kv1.3 (**B**), Kv1.4 (**C**), or Kv4.2 (**D**). (**E**,**F**) Avd3i did not inhibit voltage-gated sodium channels Nav1.5 (**E**) or Nav1.7 (**F**). The currents of these channels were evoked at 0 mV, and cell membranes were held at −80 mV.

**Figure 4 toxins-17-00461-f004:**
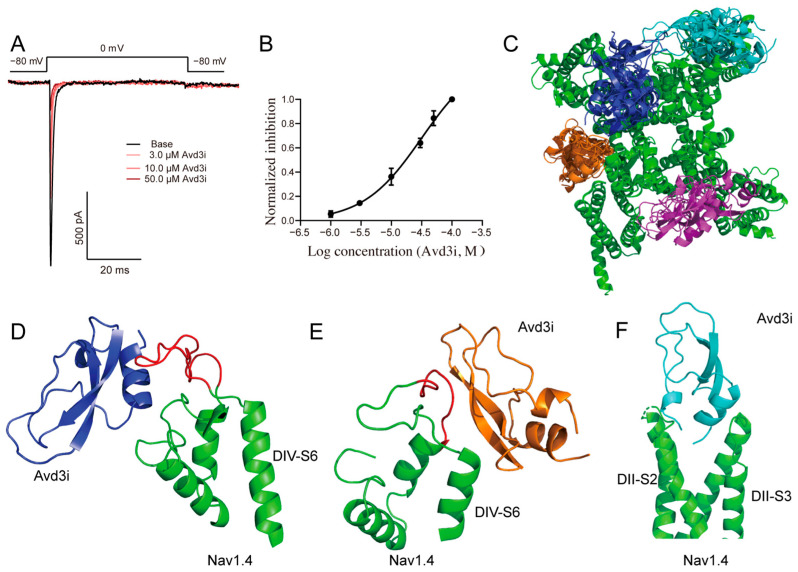
Inhibitory effect of peptide Avd3i on Nav1.4 channels. (**A**) Current traces and effect–concentration relationship (**B**) of peptide Avd3i inhibiting the Nav1.4 channel. The relationship was fitted to Hill’s equation (*n* = 4). (**C**) In the molecular docking of peptide Avd3i with the Nav1.4 channel (PDB ID: 5XSY), the Nav1.4 channel is shown in green and different clusters of Avd3i are shown in dark blue, brown, purple, and cyan, respectively. (**D**–**F**) Representative poses of Avd3i binding to the Nav1.4 channel via molecular docking. The α helix structure of Avd3i interacting with the junction between Nav1.4 DIV-S6 and DIV-S5 (**D**), one β parallel sheet in Avd3i binds to the junction between Nav1.4 DIV-S6 and DIV-S5 (**E**), and the N-terminal and C-terminal in Avd3i were inserted into the cavern formed by Nav1.4 DII-S2 and DII-S3 (**F**).

## Data Availability

The data presented in this study are available on request from the corresponding author. (Authors are going to explore further function and the data will be available on request.)

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
