# Peer review of "Sea Anemone-Derived Toxin Avd3i Inhibited Sodium Channel Nav1.4"

_toxins, 2025, doi:10.3390/toxins17090461_

Round 1
Reviewer 1 Report
Comments and Suggestions for Authors
The authors report on the expression and purification of a sea anemone U-actin toxin, which inhibits the Nav1.4 channel. This study opens up new possibilities for treating muscle diseases and contributes to the advancement of research on ion channels and natural toxins. While the research presents significant advances, several areas could be improved to enhance its impact.
Further details regarding the experimental procedures are necessary. Data from the A. viridis cDNA library or publication should be included. The E. coli strain used needs to be specified, as variants of E. coli BL21 (DE3) exist. Additionally, the full name of IPTG should be provided, along with the culture shaking speed, variables for ultrasonic crushing and centrifugation used to collect bacterial cells, as well as the conditions for TEV (including concentration, incubation times, and temperature). Details on the SDS-PAGE acrylamide concentration and gel staining method are also required, along with the column designation for C4 RP-HPLC.
Information about the sample lyophilization conditions must be included, such as the equipment used and the sample preparation process. Complete data on the HEK cell line should be provided as well.
It is important to determine the protein production yield.
In the gel presented in Figure 2B, a band of approximately 20 kDa is observed after purification with RP-HPLC. Which is the sequence of this molecule known? How might this impact the results?
While a sequence alignment is provided, a more comprehensive analysis could be achieved by performing both sequence and structural alignments with anemone toxins that target the Nav1.4 channel, including a comparison of the reported EC50 values.
Including a conclusion section could also enhance the overall impact of the work.
Furthermore, it is suggested that the following points be addressed experimentally or included in the discussion:
- Broader Functional Studies: Although Avd3i's selectivity for Nav1.4 has been established, exploring its effects on other sodium channel subtypes and in different cellular or animal models could help confirm its specificity and safety.
- Toxicity and Side Effect Assessment: Detailed studies on the toxicity of the Avd3i peptide and its potential adverse effects in biological systems are necessary, especially given its therapeutic potential.
- Validation in Animal Models: Since the research is primarily based on in vitro experiments, validating the findings in animal models would be important to evaluate the peptide's efficacy and safety in a more complex biological environment.
It is necessary to attend to the language edition
Author Response
Reviewer 1:
Comments and Suggestions for Authors
The authors report on the expression and purification of a sea anemone U-actin toxin, which inhibits the Nav1.4 channel. This study opens up new possibilities for treating muscle diseases and contributes to the advancement of research on ion channels and natural toxins. While the research presents significant advances, several areas could be improved to enhance its impact.
Thank you very much for taking the time to review our manuscript. Please find the detailed responses below and the corresponding revisions/corrections highlighted changes in the revised files.
Further details regarding the experimental procedures are necessary. Data from the A. viridis cDNA library or publication should be included. The E. coli strain used needs to be specified, as variants of E. coli BL21 (DE3) exist. Additionally, the full name of IPTG should be provided, along with the culture shaking speed, variables for ultrasonic crushing and centrifugation used to collect bacterial cells, as well as the conditions for TEV (including concentration, incubation times, and temperature). Details on the SDS-PAGE acrylamide concentration and gel staining method are also required, along with the column designation for C4 RP-HPLC.
Thank you very much. We have addressed the issues as you suggested in the method section.
Information about the sample lyophilization conditions must be included, such as the equipment used and the sample preparation process. Complete data on the HEK cell line should be provided as well.
As you suggested, we have completed the data for lyophilization conditions and HEK cells.
It is important to determine the protein production yield.
In the gel presented in Figure 2B, a band of approximately 20 kDa is observed after purification with RP-HPLC. Which is the sequence of this molecule known? How might this impact the results?
The peptide was purified with RP-HPLC in Figure 2C, showing that the peptide was obtained with high purity. Its molecular weight was identified by MALDI-TOF in Figure 2D. We have addressed the issue.
While a sequence alignment is provided, a more comprehensive analysis could be achieved by performing both sequence and structural alignments with anemone toxins that target the Nav1.4 channel, including a comparison of the reported EC50 values.
Thank you very much. Peptide Avd3i in the paper belongs to Kunitz-type peptide. So, we did its sequence alignment with some Kunitz-type peptides. Although there have been numbers of anemone neurotoxins (including ATX-II and Cgtx-II) modulating sodium channels (doi.org/10.3390/toxins15010008), it is not reported that Kunitz-type peptide target the sodium channel. we hardly found the toxins that target the Nav1.4 channel to perform the structural alignments with peptide Avd3i.
Including a conclusion section could also enhance the overall impact of the work.
Thank you very much. As you suggested, the conclusion section was included.
Furthermore, it is suggested that the following points be addressed experimentally or included in the discussion:
Broader Functional Studies: Although Avd3i's selectivity for Nav1.4 has been established, exploring its effects on other sodium channel subtypes and in different cellular or animal models could help confirm its specificity and safety.
Thank you very much. As another reviewer suggested, we cannot conclude that the peptide was selective towards Nav 1.4 channel due that the remaining VGSC subtypes were not tested. So, we revised the title as “Sea anemones-derived toxin Avd3i inhibited sodium channel Nav1.4”.
Toxicity and Side Effect Assessment: Detailed studies on the toxicity of the Avd3i peptide and its potential adverse effects in biological systems are necessary, especially given its therapeutic potential.
Validation in Animal Models: Since the research is primarily based on in vitro experiments, validating the findings in animal models would be important to evaluate the peptide's efficacy and safety in a more complex biological environment.
Considering that the peptide is not a potent inhibitor, with an IC50 of 25.43 μM. On the other hand, we also found that avd3i inhibited serine proteases at the nanomolar level (the work will be completed soon), so it is difficult to prove that the peptide may exert physiological effects by inhibiting Nav1.4. The paper focused on a novel peptide that can act on the Nav1.4 channel, which might provide a reference for the design of related targeted drugs. We are unwilling to check its physiological significance in animal model.
Reviewer 2 Report
Comments and Suggestions for Authors
Manuscript Review: Sea anemones-derived toxin Avd3i selectively inhibited sodium channel Nav1.4
General Comment:
- Title should be revised as activity in the remaining VGSC subtype has not been tested therefore it cannot be concluded that it is selective towards Nav 1.4.
- Protease activity should be tested. This peptide could have a more potent activity as a protease inhibitor than as an inhibitor of Nav1.4
- In the result section, please discuss how the Avdi3 sequence was obtained? Did the authors sequence it themselves or is it from a previously published work? Please analyze the sequence alignment with special attention to domains in Avd3i that may suggest protease or Voltage-gated ion channel activity.
- Discuss why no activity against Kv 1.1 while APEKTx1 is a potent inhibitor of the channel.
- Is this the first reported kunitz peptide with activity against VGSC? It appears that this is not a potent inhibitor but please discuss as this is important especially if indeed this is the first to have activity against VGSC.
- Please edit carefully as there are some obvious grammatical errors.
Detailed comment:
Lines 28 – 29: Please revise as all ion channels are important for human physiology. The sentence as written makes it seem that only Nav’s are important.
Line 54: Please revise as all compounds that act on the nervous system could be considered neurotoxins.
Lines 58 – 59: Please clarify what “combining” means. Do you mean they differentially bind to the voltage sensor, and they have varying degrees of affinity towards insect and mammalian Nav’s?
Lines 68-69: Which peptide is being compared to Avdi3? This sentence is vague.
Line 82: Is this a different species from A. sulcate in line 6? Quite unclear which peptide is 90%, 86% and 73% similar to Avd3i. Please rewrite this sentence. Is Ae13a same as APEKTx1? Please be consistent with names to avoid confusion.
Lines 83-84: The sentence seems out of place with no apparent relation to the previous sentences. Please rewrite.
Line 92: Are the five sequences at the bottom of Fig 1A a continuation of the sequences on top? If so please label them as well for clarity.
Line 95: Please discuss how MALDI mass analysis confirmed the correct synthesis of the peptide to explain to readers who are not familiar with this method.
Line 137: Very high suggesting very weak activity and most likely not physiologically significant. This should be reflected in the discussions.
Lines 144-146: Which of these is the most probable binding mode of Avd3i?
Lines 148-149: Why is this special? Could Avd3i inhibit other VGSC subtypes with greater potency? There are 9 and only 3 were tested. The peptide could possibly have more potent activity as a protease inhibitor. Why was this activity not tested given that the IC50 against Nav 1.4 is in the uM range?
Lines 217-219: This is quite a strong statement considering that: 1) The IC50 is high; 2) other VGSC and VGKC subtypes were not tested; and 3) the protease activity is not tested as well. Please revise this in the context of what has been done and not done in this work. I suggest not using the word selective here considering what were highlighted above.
Lines 219-222: You can't really make this statement as this peptide had not been exhaustively tested against all other channels. I suggest deleting this statement as this is too speculative with very little experimental data supporting it.
Lines 238-240: Please explain this further. Are these 4 sites occupied simultaneously by the peptide? If so, does it imply cooperative binding? Do the electrophysiological data support cooperative binding?

Please edit carefully. There are several sentences that seem out of place and the grammar can be improved.
As written the main message of the paper is communicated satisfactorily but can be improved with thorough edits.
Author Response
Reviewer 2:
Comments and Suggestions for Authors
Manuscript Review: Sea anemones-derived toxin Avd3i selectively inhibited sodium channel Nav1.4
General Comment:
Title should be revised as activity in the remaining VGSC subtype has not been tested therefore it cannot be concluded that it is selective towards Nav 1.4.
Thank you very much. As you suggested, we revised the title as “Sea anemones-derived toxin Avd3i inhibited sodium channel Nav1.4”.
Protease activity should be tested. This peptide could have a more potent activity as a protease inhibitor than as an inhibitor of Nav1.4.
Thank you for your profound comments on the manuscript. Numbers of Kunitz-type peptides was reported to inhibit serine proteases and considered to be an important and promising antithrombotic molecule. We are doing series of experiments that Avd3i suppresses serine proteases including clotting factors. However, now we are unwilling to talk about the activity of Avd3i on serine proteases. In this paper, we would like to focus on its function on ion channels.
In the result section, please discuss how the Avdi3 sequence was obtained? Did the authors sequence it themselves or is it from a previously published work? Please analyze the sequence alignment with special attention to domains in Avd3i that may suggest protease or Voltage-gated ion channel activity.
As another reviewer suggested, we explained the Avd3i sequence data in the method section. The Avd3i sequence was obtained from the cDNA library of A. viridis (NCBI Accession: PRJNA260824 ID: 260824).
Considering that we focused on the activity of Avd3i on ion channels, we labeled the probable residues (Phe13 and Arg15) important for Kv channel inhibition in Figure 1A (according to the toxin APEKTx1, doi:10.1016/j.bcp.2011.03.023).
Discuss why no activity against Kv 1.1 while APEKTx1 is a potent inhibitor of the channel.
As you suggested, we discuss the differences in the activities of these two peptides on Kv1.1 channels in the discussion section. We speculated that a substitution of Pro at residue Phe13 might be important for its inhibition on Kv1.1 channel.
Is this the first reported kunitz peptide with activity against VGSC? It appears that this is not a potent inhibitor but please discuss as this is important especially if indeed this is the first to have activity against VGSC.
Although there are no more reports that Kunitz peptide inhibited VGSC. However, considering that avd3i is not a potent inhibitor of Nav1.4 channel, we are unwilling to state the finding as “the first report”.
Please edit carefully as there are some obvious grammatical errors.
As you suggested, we addressed the issue.
Detailed comment:
Lines 28 – 29: Please revise as all ion channels are important for human physiology. The sentence as written makes it seem that only Nav’s are important.
We have addressed the issue as you suggested.
Line 54: Please revise as all compounds that act on the nervous system could be considered neurotoxins.
Thank you very much. We have addressed the issue as you suggested.
Lines 58 – 59: Please clarify what “combining” means. Do you mean they differentially bind to the voltage sensor, and they have varying degrees of affinity towards insect and mammalian Nav’s?
We have addressed the issue as you suggested.
Lines 68-69: Which peptide is being compared to Avdi3? This sentence is vague.
We have addressed the issue.
Line 82: Is this a different species from A. sulcate in line 60? Quite unclear which peptide is 90%, 86% and 73% similar to Avd3i. Please rewrite this sentence. Is Ae13a same as APEKTx1? Please be consistent with names to avoid confusion.
Thank you very much. They are the same species. And we have addressed the issue.
We have restated the sentence as “Avd3i shared a high homology with Kunitz-type peptides derived from sea anemones, such as 90%, 86% and 73% similarity with Avd3d, Avd3b[12] and APEKTx1[13], respectively”.
The toxin Ae13a is same as APEKTx1. As you suggested, we replaced its name as same.
Lines 83-84: The sentence seems out of place with no apparent relation to the previous sentences. Please rewrite.
As you suggested, we rewrite the sentence as “Textilinin-1 is a typical Kunitz-type peptide extracted from the Eastern brown snake”.
Line 92: Are the five sequences at the bottom of Fig 1A a continuation of the sequences on top? If so, please label them as well for clarity.
We have addressed the issue as you suggested.
Line 95: Please discuss how MALDI mass analysis confirmed the correct synthesis of the peptide to explain to readers who are not familiar with this method.
As you suggested, we discussed the MALDI mass analysis.
Line 137: Very high suggesting very weak activity and most likely not physiologically significant. This should be reflected in the discussions.
Thank you very much for your profound comments on the manuscript. Considering that the peptide is not a potent inhibitor, the paper focuses on avd3i as a novel acting on Nav1.4 channel, which may provide some reference for the drug design targeting at the channel. On the other hand, we also found that avd3i inhibited serine proteases at the nanomolar level (the work will be completed soon), so it is difficult to prove that the peptide may exert physiological effects by inhibiting Nav1.4.
Lines 144-146: Which of these is the most probable binding mode of Avd3i?
As shown in Figure 4C, we analyzed the various poses that avd3i might binding to Nav1.4 channel. There were four different clusters that might reflect the interaction. Considering that molecular interaction is a dynamic process, it is hard to infer the most probable binding mode simply based on the score of molecular docking.
Lines 148-149: Why is this special? Could Avd3i inhibit other VGSC subtypes with greater potency? There are 9 and only 3 were tested. The peptide could possibly have more potent activity as a protease inhibitor. Why was this activity not tested given that the IC50 against Nav 1.4 is in the uM range?
As you suggested, we addressed the issue.
Lines 217-219: This is quite a strong statement considering that: 1) The IC50 is high; 2) other VGSC and VGKC subtypes were not tested; and 3) the protease activity is not tested as well. Please revise this in the context of what has been done and not done in this work. I suggest not using the word selective here considering what were highlighted above.
Thank you very much. as you suggested, we addressed the issue.
Lines 219-222: You can't really make this statement as this peptide had not been exhaustively tested against all other channels. I suggest deleting this statement as this is too speculative with very little experimental data supporting it.
We addressed the issue, as you suggested.
Lines 238-240: Please explain this further. Are these 4 sites occupied simultaneously by the peptide? If so, does it imply cooperative binding? Do the electrophysiological data support cooperative binding?
We did not have more data supporting the cooperative binding. By molecular docking, we just analyzed the probable poses that peptide avd3i acting on Nav1.4 channel. We addressed the issue.
Round 2
Reviewer 1 Report
Comments and Suggestions for Authors
The title contradicts the conclusion. The abstract and conclusion suggest a possible inhibition, whereas the paper's title explicitly states that inhibition exists. In this regard, the title must be consistent with the conclusion of the paper.
The name Avd3i is in some places and lowercase in others; use consistent writing throughout the text.
It is suggested that centrifugation units be expressed in xg; otherwise, when expressed in rpm, the rotor specifications used must be provided.
What is the amount of total protein obtained from a specific volume of culture?
Minor language revision suggested
Author Response
Reviewer 1:
The title contradicts the conclusion. The abstract and conclusion suggest a possible inhibition, whereas the paper's title explicitly states that inhibition exists. In this regard, the title must be consistent with the conclusion of the paper.
Thank you very much. As you suggested, we revised the abstract and conclusion to be consistent with the title.
The name Avd3i is in some places and lowercase in others; use consistent writing throughout the text.
Thank you very much. We have addressed the issue.
It is suggested that centrifugation units be expressed in xg; otherwise, when expressed in rpm, the rotor specifications used must be provided.
We have addressed the issue.
What is the amount of total protein obtained from a specific volume of culture?
For the peptide Avd3i, we can get 1-2 mg per 1 L LB medium.
Comments on the Quality of English Language
Minor language revision suggested.
The English editor in MDPI helped us to undergo English language editing.
Reviewer 2 Report
Comments and Suggestions for Authors
Line 129: This should be perfused not perfumed.
Please add a little discussion on the preliminary result of the experiments testing the protease activity of this peptide since you are already testing this against serine protease.
Comments on the Quality of English LanguagePlease edit a little bit more to make the message clearer especially to readers who are not familiar with this type of research.
Author Response
Line 129: This should be perfused not perfumed.
As you suggested, we revised the sentence as “As shown in Fig.3A-D, 50 μM of Avd3i was added to HEK cells transfected with different channel plasmids.”
Please add a little discussion on the preliminary result of the experiments testing the protease activity of this peptide since you are already testing this against serine protease.
As you suggested, we add a little discussion about the peptide against serine proteases. However, we are unwilling to display the detailed function since that the paper will be finished soon.
Comments on the Quality of English Language
Please edit a little bit more to make the message clearer especially to readers who are not familiar with this type of research.
Thank you very much. The English editor in MDPI helped us to undergo English language editing.